# Surgery in Small-Cell Lung Cancer

**DOI:** 10.3390/cancers13030390

**Published:** 2021-01-21

**Authors:** Nicola Martucci, Alessandro Morabito, Antonello La Rocca, Giuseppe De Luca, Rossella De Cecio, Gerardo Botti, Giuseppe Totaro, Paolo Muto, Carmine Picone, Giovanna Esposito, Nicola Normanno, Carmine La Manna

**Affiliations:** 1Thoracic Surgery, Istituto Nazionale Tumori, “Fondazione G. Pascale”—IRCCS, 80131 Naples, Italy; a.larocca@istitutotumori.na.it (A.L.R.); g.deluca@istitutotumori.na.it (G.D.L.); c.lamanna@istitutotumori.na.it (C.L.M.); 2Thoracic Medical Oncology, Istituto Nazionale Tumori, IRCCS “Fondazione G. Pascale”, 80131 Naples, Italy; a.morabito@istitutotumori.na.it (A.M.); espositogiovanna87@gmail.com (G.E.); 3Pathology, Istituto Nazionale Tumori, “Fondazione G. Pascale”–IRCCS, 80131 Naples, Italy; r.dececio@istitutotumori.na.it; 4Scientific Directorate, Istituto Nazionale Tumori, “Fondazione G. Pascale”–IRCCS, 80131 Naples, Italy; g.botti@istitutotumori.na.it; 5Radiotherapy, Istituto Nazionale Tumori “Fondazione G. Pascale”–IRCCS, 80131 Naples, Italy; g.totaro@istitutotumori.na.it (G.T.); p.muto@istitutotumori.na.it (P.M.); 6Radiology, Istituto Nazionale Tumori, “Fondazione G. Pascale”–IRCCS, 80131 Naples, Italy; c.picone@istitutotumori.na.it; 7Cellular Biology and Biotherapy, Istituto Nazionale Tumori, “Fondazione G. Pascale”—IRCCS, 80131 Naples, Italy; n.normanno@istitutotumori.na.it

**Keywords:** small-cell lung cancer, lobectomy, pneumonectomy, radiotherapy, chemotherapy, multimodal treatment

## Abstract

**Simple Summary:**

Small-cell lung cancer (SCLC) accounts for approximately 15% of all lung cancers and is one of the most aggressive tumors, with poor prognosis and limited therapeutic options. This review summarizes the main results observed with surgery in SCLC, discussing the critical issues related to the use of this approach. Following two old randomized clinical trials showing no benefit with surgery, several prospective, retrospective, and population-based studies have demonstrated the feasibility of a multimodality approach including surgery in addition to chemotherapy and radiotherapy in patients with selected stage I SCLC. Currently, the International Guidelines recommend a surgical approach in selected stage I SCLC patients, after adequate staging within a multimodal approach and after a multidisciplinary evaluation.

**Abstract:**

Small-cell lung cancer (SCLC) is one of the most aggressive tumors, with a rapid growth and early metastases. Approximately 5% of SCLC patients present with early-stage disease (T1,2 N0M0): these patients have a better prognosis, with a 5-year survival up to 50%. Two randomized phase III studies conducted in the 1960s and the 1980s reported negative results with surgery in SCLC patients with early-stage disease and, thereafter, surgery has been largely discouraged. Instead, several subsequent prospective studies have demonstrated the feasibility of a multimodality approach including surgery before or after chemotherapy and followed in most studies by thoracic radiotherapy, with a 5-year survival probability of 36–63% for patients with completely resected stage I SCLC. These results were substantially confirmed by retrospective studies and by large, population-based studies, conducted in the last 40 years, showing the benefit of surgery, particularly lobectomy, in selected patients with early-stage SCLC. On these bases, the International Guidelines recommend a surgical approach in selected stage I SCLC patients, after adequate staging: in these cases, lobectomy with mediastinal lymphadenectomy is considered the standard approach. In all cases, surgery can be offered only as part of a multimodal treatment, which includes chemotherapy with or without radiotherapy and after a proper multidisciplinary evaluation.

## 1. Introduction

Small-cell lung cancer (SCLC) accounts for approximately 15% of all lung cancers and it is one of the most aggressive tumors, with a rapid growth and early metastases [1,2]. It is typically associated with tobacco use (90% of cases): the risk of developing the disease increases with the duration of smoking and the number of cigarettes smoked each day. The traditional staging system has been developed by the Veteran’s Administration Lung Cancer Study Group (VALSG) in the ’50s in the United States and it classifies SCLC according to the extent of disease into two stages, extensive and limited [3]. Extensive stage (ES)-SCLC extends one radiation portal, including distant metastases and malignant pleural effusions: it is diagnosed in approximately 70% of patients and has a poor prognosis, with a median survival of about 10–12 months and only about 2% of patients surviving for 5 years [4]. Limited-stage (LS)-SCLC is confined within one radiation portal, defined as a single hemithorax with ipsilateral and supraclavicular nodes: it is diagnosed in approximately 30% of patients and presents a more favorable outcome, with a median survival of 15–20 months, 2- and 5-year survival rates of 20–40% and 12–25%, respectively [5,6]. Moreover, approximately 5% of patients present with early-stage SCLC (T1,2 N0,M0): these patients have a better prognosis, with a 5-year survival up to 50% [7,8,9]. In this group of patients, a surgical approach can be proposed as part of a multidisciplinary treatment after excluding mediastinal lymph nodes involvement, according to the National Comprehensive Cancer Network (NCCN) guidelines [10]. This review summarizes the main results observed with surgery as single treatment or as part of a multimodality treatment of SCLC, discussing the critical issues related to the use of this approach. We proceeded to a revision of the Medline PUBMED English literature (from January 1959 to December 2020) and we grouped the studies found according to the design (randomized, prospective non-randomized, retrospective and cancer data-based review), with the objective to verify the impact of surgery on survival (reported as median survival and 5-year survival) of patients with SCLC.

## 2. Surgery as Single Treatment for SCLC

Before the 1960s, surgery has been the treatment of choice for resectable SCLC cases. In 1959, Belcher et al. reported a 5-year survival of 37% in 42 SCLC patients treated with pulmonary lobectomy [11]. In 1962, in a large series of 386 SCLC patients of the Memorial Hospital for Cancer and Allied Diseases of New York, N.Y., USA Watson et al. reported that surgical resections were performed in 7% of SCLC patients, including pneumonectomy in 67%, lobectomy in 22% and wedge resection in 11% of cases [12]. However, only 11% of resected patients have survived for more than 4 years. An exploratory thoracotomy revealed a non-resectable tumors in 84 cases (22% of patients), treated subsequently with different palliative therapies. With radiation therapy alone, only 1 patient survived more than 2 years; 90% of patients died in less than 1 year. In 1960s, the Medical Research Council conducted a randomized trial comparing surgery versus radical radiotherapy in patients with SCLC, without extrathoracic metastases, considered to be operable and fit enough for radical radiotherapy (Table 1) [13]. Overall, 144 patients were admitted to the trial from 29 thoracic centers throughout Great Britain: 71 patients were randomized to surgery arm and 73 to radiotherapy arm. Among patients randomized to the surgical arm, only 48% underwent a complete resection: an explorative thoracotomy was performed in 34% of cases and surgery was definitively excluded in 18% of patients. Among patients randomized to radiotherapy arm, 85% of patients received a curative treatment, while 11% of patients received only a palliative treatment and 4% no radiotherapy at all. The ten-year follow-up of this trial was published in 1973 and showed that the median survival for the surgical arm was 199 days versus 300 days of the radiotherapy arm (*p* = 0.04). Patients who received curative radiotherapy had a higher survival rate than those undergoing surgery over the 2-year (11% vs. 6%) and 5-year (5% vs. 0) period. Moreover, there were no 10-year survivors in the surgical series, while 3 patients remained alive in the radiotherapy group. These results reinforced the previous conclusions of the 2- and 5-year reports of this trial, suggesting that radical radiotherapy was superior to surgery in terms of overall survival in patients with limited SCLC judged to be operable [14,15]. On the basis on these data, the role of surgery alone in limited SCLC gradually decreased and a combined modality of treatment including chemotherapy and radiotherapy (better if starting within 30 days after the beginning of chemotherapy) became the cornerstone of treatment of patients with limited-stage SCLC [16,17,18,19,20]. The combined treatment reduced, in particular, the risk of a thoracic recurrence, while brain metastases became one of the main types of relapse, leading to several trials that evaluated the role of prophylactic cranial irradiation in patients with SCLC with contrasting results [21,22,23]. In 1999, the meta-analysis of Auperin A. et al. confirmed the role of prophylactic cranial irradiation in reducing the risk of brain metastases and improving overall and disease-free survival of SCLC patients with limited disease [24]. Therefore, the combination of chemotherapy (cisplatin and etoposide) plus chest radiotherapy followed by prophylactic cranial irradiation has been considered the standard treatment for patients with limited-stage SCLC and good performance status, with an objective response rate of approximately 80%, a median overall survival of about 17 months and 12–25% of patient cancer-free at 5 year [4,25].

## 3. Surgery Plus Chemotherapy/Radiotherapy in SCLC

### 3.1. Randomized Studies

The role of surgery to the multimodality management of SCLC has been evaluated by a large multicenter randomized phase III trial, promoted in 1983 by the Lung Cancer Study Group (LCSG), that became an intergroup study with the participation of the Eastern Cooperative Oncology Group (ECOG) and the European organization for Research and Treatment of Cancer (EORTC) (Table 1) [26]. Overall, 328 patients with LS-SCLC were enrolled into the study and treated with cyclophosphamide, doxorubicin, and vincristine for five cycles. Patients who achieved at least a partial response and who were fit enough for surgery were randomized to undergo or not to undergo pulmonary resection and all randomized patients were treated with chest and brain radiotherapy. Among 217 responders, only 146 patients (66%) were randomized to surgery (70 patients) and to no surgery (76 patients). The resection rate was 83%: a complete resection was feasible in 77% of patients and 19% patients had a pathologic complete response. No difference in overall survival was observed between the surgical and no surgical arms (median OS: 15.4 vs. 18.6 months, respectively; *p* = 0.78). Two-year survival was 20% in both groups. Therefore, this trial does not support the efficacy of the addition of pulmonary resection to the multimodality treatment of SCLC. Limits of the trial are the number of incomplete resection (23%), the lack of platinum-based chemotherapy in the neoadjuvant phase of treatment, the possible understadiation of patients due to the unavailability of positron emission tomography (PET)scan. However, based on the results of this trial, surgery has been largely discouraged, also as part of a multimodality strategy of treatment of patients with limited SCLC.

### 3.2. Prospectives Studies

In the following years, several prospective non-randomized trials have reevaluated the role of surgery in selected LS-SCLC patients (Table 2). A prospective study of adjuvant surgical resection after chemotherapy for patients with LS-SCLC has been conducted by the University of Toronto Lung Oncology Group and published by Shepherd F. et al., on 1989 [27]. Overall, 72 patients received preoperative chemotherapy (cyclophosphamide, doxorubicin and vincristine or cisplatin and etoposide): 80% of patients had an objective response, 79.1% were considerate eligible for surgery, but only 38 patients underwent thoracotomy. The median survival for the resected patients was 21 months and the 5-year survival rate was 36%. A larger experience of the same group on the multimodality treatment of SCLC with surgery and chemotherapy was reported by Shepherd F. et al., on 1991 and it included 119 patients with LS-SCLC [28]. Seventy-nine patients had surgery first followed in 67 cases by adjuvant chemotherapy, while 40 patients had chemotherapy first, followed by surgery. The 5-year survival rate for the whole population was 39%; no difference in terms of overall survival was seen between the two groups of patients (*p* = 0.756). Patients with pathologic stage I had a 5-year survival rate of 51%, significantly better than patients in stage II (28%, *p* = 0.001) and III (19%, *p* = 0.001), supporting the evidence in favor of surgery for patients with stage I disease. Similar results were reported on 1995 by Karrer K. et al., for the LCSG of the International Society of Chemotherapy, in a prospective trial for patients with early-stage SCLC (T1,2,N0,M0) [29]. A total of 183 patients received surgery, followed by 8 cycles of standard chemotherapy (cyclophosphamide, doxorubicin and vincristine) or 6 intermittent cycles of alternating chemotherapy with 3 different drug combinations, and thereafter by prophylactic cranial irradiation. Overall, 152 patients (83%) had a complete resection, resulting in a 3-year survival rate of 44%, while it was 19% for patients with incompletely resected. The 4-year survival probability was 57% for 68 patients with stage pT1-2N0M0R0 after complete resection and it was 37% for patients with stage pT1-2N2M0R0 after surgery. In 1997, another pilot phase II study evaluated the feasibility and activity of a multimodality approach based on neoadjuvant chemotherapy followed by surgical resection in 22 Japanese SCLC patients with stage I–IIIA [30]. All patients received 2–4 cycles of neoadjuvant CAV II (cisplatin, doxorubicin, etoposide), with a response rate of 95.5%, and 21 patients underwent a surgical resection. Median survival was 61.9 months, and the 3-year survival probability was 66.7%, higher for patients with stage I and II than for patients with stage III (73.3% versus 42.9%, respectively, *p* = 0.018). One operation-related death occurred. In 1998 Rea F. et al. reported the results of a large Italian prospective study on 104 patients with SCLC (49% in stage I–II and 51% in stage III) treated at the University of Padua, Italy from 1981 to 1995 with surgery followed by adjuvant chemotherapy and radiotherapy (stage I–II) or with induction chemotherapy followed by surgery and radiotherapy (stage III) [31]. The 30-day mortality was 2%. Median overall survival was 28 months, and the 5-year survival rate was 32%: according to the pathologic stage, 5-year survival was 52.2%, 30% and 15.3% for stage I, II, and III, respectively (*p* < 0.001). In patients without residual tumor after chemotherapy and surgery, the 5-year survival rate was 41%.

In 1999 Eberhardt W. et al. reported the results of multimodality approach including surgery, chemotherapy and radiotherapy in German patients with stage IA–IIIB SCLC [32]. In stage IB/IIA patients received four cycles of cisplatin and etoposide followed by surgery; in stage IIB/IIIA patients received three cycles of cisplatin and etoposide followed by a concurrent chemoradiation cycle including hyperfractionated accelerated radiotherapy and surgery. Forty-six consecutive SCLC patients were enrolled in this study: 6 in stage IB, 2 in stage IIA, 22 in stage IIB/IIIA and 16 in stage IIIB. Forty-three patients (94%) showed an objective response and 23 (72%) underwent radical surgery (R0): 6 patients in stage IB, 2 in stage IIA, 13 in stage IIB/IIIA and 2 in stage IIIB. No perioperative deaths occurred, but a patient died of septicemia. Median survival was 36 months for all patients and 68 months in R0 patients. The 5-year survival rate was 46% and 63% for all patients and for R0 patients, respectively. The authors concluded that this multimodality treatment including surgery, chemotherapy, and radiotherapy proved highly effective with high local control and remarkable long-term survival after complete resection, even in SCLC patients with stages IIB/IIIA. In 2005, the Japan Clinical Oncology LCSG published the results of a phase II trial to determine the feasibility and activity of lung resection followed by adjuvant chemotherapy for SCLC patients with stage I–IIIa [33]. Sixty-two patients with completely resected SCLC entered in the trial and 69% received 4 cycles of cisplatin and etoposide. No treatment associated mortality was observed. Three-year survival was 68% in patients with stage I, 56% in stage II, and 13% in stage IIIa (*p* = 0.02). Local failure was observed in 10% of patients, less frequently in patients with stage IA (4%) and more frequently in patients with stage IIIA (22%). Therefore, also this trial confirmed the feasibility and the good outcome of a surgical approach followed by adjuvant chemotherapy: however, considering that nodal metastases are a major prognostic factor, the authors highlighted the importance of a preoperative evaluation of mediastinal nodal status.

### 3.3. Retrospective Studies

The evidence coming from prospective non-randomized studies have been confirmed by retrospective studies that have evaluated the role of surgery in LS-SCLC patients (Table 3). The role of initial surgical resection in patients in patients with SCLC has been retrospectively evaluated by the Veterans Administration Surgical Oncology Group and published by Shields et al., on 1982 [34]. The potentially “curative” resections represented 4.7% of all “curative” resections carried out in four prospective adjuvant chemotherapy trials. In the 132 patients included in the analysis, the 5-year survival was 23%, but it was 60%, 31% and 28% in patients with stage pT1N0, pT1N1, and pT2N0, respectively. They concluded that resection is indicated in patients with early disease pT1NO, probably indicated for those with pT2N0 or pT1N1 and contraindicated in patients with any other TNM category.

The long-term benefit of a multimodality strategy based on surgery followed by adjuvant chemotherapy has been reported by Osterlind K. et al., in 1986, analyzing a consecutive series of 874 SCLC patients treated with chemotherapy with or without radiotherapy at the Finsen Institute, Copenhagen, Denmark, between 1973 and 1981 [35]. Among 437 patients with limited disease, 150 were considered operable: 52 patients underwent a radical resection, 44 were considered non-resectable at the thoracotomy and 54 operable patients were not operated due to the treatment policy at the hospitals from which they were referred that excluded surgery for SCLC. Overall, 36 patients received a radical resection, while 16 patients had microscopic (9 cases) or macroscopic (7 cases) residual tumor. The 30-month disease-free-survival (DFS) rate was 33% for completely resected patients, 12.5% for those with residual disease and 13% for patients operable but not operated, suggesting a possible role for surgery in limited SCLC patients with early disease.

The role of surgery in the treatment of 81 Japanese patients with clinically localized SCLC was evaluated by Hara N. et al. [36]. Overall, 36 patients underwent surgical resection: the surgery was done upfront in 19 cases followed by adjuvant chemotherapy and after neoadjuvant chemotherapy in 17 cases. The remaining 45 patients received chemotherapy plus radiotherapy. Median OS was 33 months, and the 5-year survival was 38% for the 36 surgical patients. Patients with stage I and II showed a 5-year survival of 25% and a median OS of 33 months.

Similar findings were reported in 2000 in another Japanese study by Inoue M. et al., on 91 SCLC patients treated with a multimodality strategy including pulmonary resection [37]. The five-year survival probability was 37.1%: it was 56.1% for stage IA, 30% for stage IB, 57.1% for stage IIA and 42.9% for stage IIB. Patients treated with surgery plus chemotherapy had a better 5-year probability of survival than that of those treated with surgery alone (54.9% versus 22.2%, respectively; *p* = 0.015). Moreover, the 5-year survival rate of patients treated with four or more cycles of chemotherapy was 80.0%. The authors concluded that thoracic resection in combination with chemotherapy treatment offers the best results in patients with stage IA-IIB SCLC. In 2004, Badzio A. et al. reported a retrospective comparative analysis of survival in 134 patients with LS-SCLC treated between 1984 and 1996 with either complete surgical resection followed by chemotherapy (67 patients), or with conventional non-surgical management (67 patients) [38]. The control group was selected using the methodology of "pair-matched case-control", among 176 patients with LS-SCLC treated without surgery, but potentially eligible for resection. The two groups were balanced for prognostic factors. Patients treated with surgery and adjuvant chemotherapy had a better median survival than those treated without surgery (22 versus 11 months, *p* < 0.001) and a lower incidence of local relapse (15% versus 55%, respectively, *p* < 0.001). The 5-year survival probabilities were 27% and 4% in the surgical and non-surgical group, respectively, suggesting a possible role of surgery in limited-stage SCLC.

These positive findings were confirmed by a large retrospective study of Brock V et al. conducted on 1415 patients with SCLC treated from 1976 to 2002 at the Johns Hopkins Medical Institutions, Baltimore, Md (USA): 82 patients (6%) had undergone curative surgery and had a 5-year survival of 42% [39]. In particular, 9/82 patients (11%) received surgery alone, 18/82 neoadjuvant chemotherapy followed by surgery (22%) and 45/82 surgery followed by adjuvant chemotherapy (55%). Prophylactic cranial irradiation was given to 23% of patients. The 5-year survival was better for patients receiving platinum versus non-platinum regimens (68% versus 32.2%, *p* = 0.04) and for patients undergoing lobectomies than limited resections (50% versus 20%, *p* = 0.03). Furthermore, the 5-year survivals for patients with stage I disease who received adjuvant platinum versus non-platinum chemotherapy were 86% versus 42%, respectively (*p* = 0.02), supporting a reconsideration of the role of surgery in the multimodality strategy of treatment for selected patients with LS-SCLC. More recently, Takenaka T. et al. compared the outcomes of surgical resections to other conventional non-surgical treatments in 277 Japanese patients who received treatment for LD-SCLC (18% with stage I) from 1974 through 2011 [40]. Surgery was performed in 31.7% of cases and included pneumonectomy in 11.1% of cases, lobectomy in 84.1% of cases and limited resections in 4.5% of cases. The 5-year survival rates for all patients according to stage were 58% in stage I, 29% in stage II and 18% in stage III. The 5-year survival rates of the patients with and without surgery were 62% and 25% in stage I (*p* < 0.01), 33% and 24% in stage II (*p* = 0.95), 18% and 18% in stage III (*p* = 0.35), respectively. Moreover, the study showed that the 5-year survival rates according to the treatment period were 20% in the 1970/1980s, 21% in the 1990s and 40% in the 2000s (*p* < 0.01). Therefore, also this study suggest that surgery is effective for patients with stage I SCLC. A Chinese retrospective study has been recently published on 2020 to analyze the effects of radical surgery and concurrent chemoradiotherapy on the prognosis of 157 patients with LS-SCLC, treated in a single Institution from 2011 to 2018 [41]. Overall, 50 patients received surgery after neoadjuvant chemotherapy or followed by adjuvant chemotherapy, while 102 patients received concurrent chemoradiotherapy. Median progression-free survival (73 versus 10.5 months, *p* < 0.0001) and overall survival (79 versus 23 months, *p* < 0.0001) were significantly longer in the surgical group than non-surgical group, respectively. Finally, a retrospective analysis has been recently published by Casiraghi M et al., reporting the outcomes of Italian patients with SCLC undergoing surgery at the European Institute of Oncology of Milan [42]. Among 324 patients treated between 1998 and 2018, 65 patients (20%) underwent surgical resection with curative intent: upfront in 60% of cases, after chemotherapy (36.9%) and after chemotherapy plus radiotherapy (3.1%). Forty-four patients (67.7%) underwent adjuvant treatment and 23.1% patients prophylactic cranial irradiation. Median overall survival after resection was 36 months, while 5 and 10-year OS was 42% and 25.4%, respectively. At multivariate analysis pathological stage was the strongest prognostic factor: in particular, *p*-stage I patients had a 5-year OS of 76.6 % (log-rank *p* < 0.0001).

## 4. Cancer Database Review

Data on larger populations with SCLC treated with surgery have been reported by studies that analyzed different national cancer data base (Table 4). One of the first cancer data base review has been published by Rostad et al., on 2004 and it was conducted on the Norway Cancer Registry, evaluating all patients with SCLC diagnosed between 1993–1999 in Norway [43]. The purpose of the study was to identify the proportion of patients with operable SCLC and to compare the resection specimens from operated patients with more than 5-year survival with those with shorter survival. Overall, 2442 patients with SCLC were identified: 697 patients were considered to have limited disease and 180 patients were classified as stage I. For stage I, 96 patients were considered potentially operable and 38 patients were effectively resected (39%): the 5-year survival rate was 11.3% in conventionally treated patients compared to 44.9% for those who underwent surgical resection. Therefore, the authors concluded that patients with SCLC in stage I could be referred to surgery as long-time survival is good. A large U.S population-based database, the Surveillance, Epidemiology, and End Results (SEER) registry, was used by Schreiber et al. to determine survival outcomes of SCLC patients who underwent surgery between 1988 and 2002, coded as localized disease (T1-T2Nx-N0) or regional disease (T3-T4Nx-N0) [8]. In total, 14,179 patients were identified in SEER registry and 863 underwent surgery. Surgery was more commonly associated with early-stage (T1T2) disease (*p* < 0.001) and with improved survival for the whole cohort (28 versus 13 months for no surgery; 5-year OS rate 34.6 versus 9.9%, *p* < 0.001), and for patients with early (42 versus 15 months; 5-year OS rate 44.8 versus 13,7%, respectively, *p* < 0.001) and regional disease (22 versus 12 months; 5-year OS rate 26.3% versus 9.3%, respectively, *p* < 0.001). Patients with early disease who underwent lobectomy had a median survival of 65 months and a 5-year OS rate of 52.6%. The multivariate analysis confirmed the benefit of lobectomy across all time intervals (*p* = 0.002). In conclusion, this population-based study confirmed the role of surgery, particularly lobectomy, in selected patients with early-stage SCLC. The same SEER database was then used by Yu et al. to better characterize outcomes of patients with SCLC in stage I treated from 1988 to 2004 [7]. A total of 1560 patients were identified: 15.8% underwent lobectomy, 7.8% a surgical resection less extensive than a lobectomy and 0.6% a pneumonectomy. Among the patients who underwent a lobectomy, 17% received chest radiotherapy. For all patients, 3- and 5-year OS was 31% and 21.1%, respectively. The 3- and 5-year OS probability was 58% and 50%, respectively, for all patients who had a lobectomy (64.9% and 57.1% for those who did also receive radiotherapy). Therefore, based on this analysis on a large series of stage I SCLC patients, surgery without radiotherapy seemed to offer good outcome in selected patients who undergo lobectomy and who are node-negative. Similar findings were reported also by Varlotto et al. who evaluated the incidence of stage I-II SCLC and defined the optimal local therapy through an analysis of 2214 early-stage SCLC patients identified in the SEER database from 1988 to 2005 [44]. Early-stage SCLC represented a 3–5% of all SCLC until 2003 and, by 2005, increased to 7%. Surgery for early-stage SCLC achieved a peak at 47% in 1990, but then progressively declined to 16% by 2005. The median OS for all patients was 20 months. Patients treated with lobectomy had longer median survival than those treated with radiotherapy alone (50 vs. 20 months, *p* < 0.0001). The use of radiotherapy did not affect prognosis after limited resection (30 vs. 28 months, *p* = 0.585). Results of multivariate analysis demonstrated that survival was independently related to age, year of diagnosis, tumor size, stage, and treatment (lobectomy versus sub-lobar resection versus radiotherapy alone). Therefore, the authors concluded that in patients with early-stage SCLC lobectomy provided superior survival, but the addition of radiotherapy to resection was associated with no additional benefit. Weksler B. et al. queried the SEER database for patients with SCLC from 1988 to 2007 and they identified 3566 patients with stage I (75.3%) o II (24.7%) [45]. Overall median survival for all patients was 18.0 months. Lung resection was performed in 25.1% of cases: median OS was 34 versus 16 months for surgical versus non-surgical patients, respectively (*p* < 0.001). Median survival was longer after lobectomy or pneumonectomy than after wedge resection (39 versus 28 months, respectively, *p* = 0.0001). Radiotherapy was performed in 49.6% of cases and in 22. 6% of resected patients. The multivariate analysis showed that female, younger age, stage I, treatment with radiotherapy, lymph node staging, and lung resection were significantly associated with survival. The analysis of the largest database on SCLC has been published by Gaspar et al., on 2012 [46]. Overall, 68,611 patients with SCLC in the National Cancer Data Base (NCDB) were analyzed to describe demographic characteristics, treatment strategies and survival changes between 1992 and 2007: 25,499 cases presented LS-SCLC. Four patient cohorts of patients diagnosed in 1992, 1997, 2002, and 2007 were examined. Median OS for patients with ES-SCLC and LS-SCLC was 6.1 and 12.9 months, respectively, and it was not significantly improved between 1992 and 2002, despite changes in demographics and treatments. Surgery alone or in combination with chemotherapy or radiotherapy was performed in 5.5% of cases and was associated with improved survival: median OS for patients with early-stage SCLC undergoing surgery or no surgery was 30.8 versus 15 months (*p* < 0.01). If surgery was performed, patients with early-stage disease benefited from the addition of chemotherapy. The multivariate analysis confirmed that female sex, age <70 years, and receipt of surgery were associated with improved survival for LS-SCLC. Radiotherapy decreased the hazard ratio for stage III SCLC patients, but not for those with earlier disease. Chemotherapy decreased the Hazard ratio (HR) for all patients with LS-CLC. Patients with ES-SCLC treated with radiation in addition to chemotherapy had better survival than those who received only chemotherapy. A retrospective analysis of 243 patients from Japanese Lung Cancer Registry who underwent surgery in 2004 has been reported by Takei H. et al., on 2014 [47]. The authors found that of the 11,663 resected patients, 243 patients had a SCLC (2.1%): the 5-year survival rate for all cases was 52.6% (64.3% in patients with stage IA). The multivariate analysis showed that the age, gender, c-stage, and surgical curability were significant prognostic factors. More recently, the NCDB was reviewed for patients with clinical T1–3 N1 M0 SCLC who underwent concurrent chemoradiation versus surgery and adjuvant therapy from 2003 to 2011 [48]. Overall, 1041 patients met the inclusion criteria: 96 patients (9%) underwent surgery and adjuvant chemotherapy with or without radiotherapy, while 945 patients (91%) underwent chemoradiotherapy alone. The 5-year survival was 31.4% for the surgery group and 26.3% for the chemoradiation group (*p* = 0.03). The multivariate analysis demonstrated that surgery plus adjuvant chemotherapy with or without radiotherapy was associated with improved survival compared with chemoradiotherapy (HR 0.74, 95%CI: 0.56 to 0.97). An improved long-term survival was observed for surgery and adjuvant chemotherapy compared with chemoradiotherapy also when the analysis was limited to 2301 node-negative SCLC patients of the same (NCDB) data base, with a 5-year OS of 47.6% versus 29.8%, *p* < 0.001 [49]. Similar findings were reported on a larger number of patients of the NCDB reviewed from 1998 to 2011. Surgery was performed in 9% of patients with potentially resectable SCLC: 5-year OS was 51%, 25%, and 18% for resected patients with stages I, II, and IIIA, respectively [50]. Addition of surgery to chemotherapy was associated with improved survival, independently of age, stage, and comorbidity score (HR: 0.57; 95%CI: 0.47–0.68). Therefore, all these studies supported the re-evaluation of the role of surgery in the multimodality treatment of early-stage SCLC patients.

## 5. Discussion and Conclusions

The scenario of treatment of SCLC is changed in the last few years, after decades of no progress. Immunotherapy with atezolizumab or durvalumab has been approved in combination with platinum and etoposide in the first line therapy of patients with ES-SCLC, while nivolumab and pembrolizumab as single agents showed anti-tumor activity and were approved in patients with ES-SCLC after platinum-based therapy and at least one prior line of therapy [51]. Moreover, recently the Food and Drug Administration granted accelerated approval to lurbinectedin, a selective inhibitor of oncogenic transcription, for patients with metastatic SCLC in progression on or after platinum-based chemotherapy, based on the positive result of a phase 2 study [52]. For patients with LS-SCLC several phase 3 studies are evaluating the role of immunotherapy in combination with chemotherapy and radiotherapy. Is there still a role for surgery in selected patients with limited-stage disease? Unfortunately, clinical evidence coming from literature is quite weak. Only two randomized clinical trials have evaluated the role of surgery in patients with SCLC, but they were both old studies, including a limited number of patients (about 150). Most evidence comes from prospective non-randomized studies and retrospective analysis, but they were generally conducted in one Institution and once again on few patients (in most cases less than one hundred). The greater evidence is derived from large cancer data base review that reported results observed in national, large series of patients (thousands or tens of thousands). The randomized trials of the MRC in the 1960s and of the LCSG in the 1980s showed the inferiority of surgery versus curative radiotherapy in terms of overall survival in patients with limited SCLC judged to be operable, previously not treated or treated with neoadjuvant chemotherapy, according to the design of MRC and LCSG study, respectively. However, limits of these old studies are the lack of chemotherapy in the MRC or the lack of a platinum-based chemotherapy in LCSG in the neoadjuvant phase, the low power and the high rate of incomplete resection of both studies, the unavailability of PET scan, with possible under-staging of patients. Several subsequent prospective studies have confirmed the feasibility of a multimodality approach including surgery before or after chemotherapy and followed in most studies by thoracic radiotherapy, with a 5-year survival probability of 36–63% for patients with completely resected stage I. These results were substantially confirmed by the retrospective studies conducted in the last 40 years: notwithstanding the limits of all these studies (retrospective evaluation, selection bias, heterogeneity of patients and treatments), the best outcome was observed for patients with limited disease who underwent surgical resection in addition to chemotherapy and radiotherapy, with a 5-year survival probability of 27–86%. Large, population-base studies, conducted from the 1990s in Europe, U.S and Japan confirmed the benefit of surgery, particularly lobectomy, in selected patients with early-stage SCLC: the resection rate was 5–9% in the different series and the 5-year survival rate was 31–51% for patients with stage I who underwent to surgical resection within a multimodality treatment strategy. Multivariate analyses confirmed, in particular, the benefit of lobectomy in early-stage SCLC, while for radiotherapy the benefit was mainly limited to stage III. In 2015 Stish et al. reviewed the outcomes and patterns of failure for 54 patients with SCLC treated with definitive surgical resection at Majo Clinic (Rochester, USA). Patients undergoing wedge resection or segmentectomy had an increased risk of intrathoracic recurrence compared with those who received a lobectomy or pneumonectomy (HR:3.5; *p* = 0.01) [53]. Moreover, the 5-year overall survival was significantly longer after lobectomy or pneumonectomy versus wedge resection or segmentectomy (48% versus 15%, respectively; *p* = 0.03).

The role of surgery in stage I-III SCLC has been also evaluated by a recent meta-analysis including two randomized trials and 13 retrospective studies for a total of 41,483 eligible patients [54]. The results of this meta-analysis confirmed that surgery significantly improved overall survival when compared to non-surgical treatments in the retrospective studies (HR: 0.56; 95%CI: 0.49–0.64, *p* < 0.001), but not in the 2 “older” randomized clinical trials (HR: 0.77; 95%CI: 0.32–1.84, *p* = 0.55). Moreover, sub-lobar resections resulted in a worse survival than lobectomy (HR: 0.64; 95%CI: 0.56–0.74, *p* < 0.001). Based on this evidence, the NCCN Guidelines Version 1.2021, the ACCP, ASCO and ESMO guidelines highlight that surgery is justified for selected stage I (T1-2,N0M0) SCLC patients [10,55,56,57]. The preferred surgical approach is lobectomy with mediastinal lymph node dissection that can be proposed after excluding a mediastinal lymph node involvement (with CT scan, PET-CT scan, or EBUS and/or mediastinoscopy if enlarged). Pathologic mediastinal staging is not required if the patient is not a candidate for surgical resection. In patients potentially eligible for surgery, staging procedures should be completed quickly, without significantly delaying the treatment, due to the aggressiveness of the disease. After surgery, adjuvant chemotherapy with platinum and etoposide for four cycles should be administered. In case of unforeseen nodal mediastinal involvement (N1 or N2) or in those patients without a systematic nodal dissection, thoracic radiotherapy after surgery should be considered. On the contrary, there is no role for surgery after induction chemotherapy in patients with N2 disease. Alternatively, due to the lack of randomized trials, combined concurrent chemoradiotherapy can be offered to patients with T1,2N0M0 and it is the first option for patients with significant concomitant medical illnesses who are at increased risk of perioperative complications. All patients with T1-2N0M0 should be considered for prophylactic cranial irradiation (PCI) after surgery and adjuvant chemotherapy.

In conclusion, the role of surgery in SCLC has been much debated and the International Guidelines recommend a surgical approach only in selected stage I patients, after adequate staging. For patient candidates for surgery, lobectomy with mediastinal lymphadenectomy is considered the standard approach, while sub-lobar resections are not considered appropriate. In all cases, surgery can be offered only as part of a multimodal treatment, which includes chemotherapy with or without radiotherapy and after a proper multidisciplinary evaluation.

## Figures and Tables

**Table 1 cancers-13-00390-t001:** Randomized trials evaluating the role of surgery.

Author	Patients	Treatment	Complete Resection Rate (%)	Median OS (Months)	2-Year OS	5-Year OS
Fox W. et al., 1973 [13]	144	Surgery vs. radiotherapy	48%	6.5 vs. 9.8, *p* = 0.04	4 vs. 10%	1 vs. 5%
Lad T. et al., 1994 [26]	146	Chemotherapy * followed by radiotherapy or surgery plus radiotherapy	77%	18.6 vs. 15.4, *p* = 0.78	20% vs. 20%	n.r.

* cyclophosphamide, doxorubicin, and vincristine for five cycles; OS: overall survival; n.r.: not reported.

**Table 2 cancers-13-00390-t002:** Prospective non-randomized trials evaluating the role of surgery.

Author	Patients	Treatment	Complete Resection Rate (%)	Median OS (Months)	5-Year OS
Shepherd F. et al., 1989 [27]	72	Chemotherapy * followed by surgery	52.7%	21	36%
Shepherd F. et al., 1991 [28]	119	Surgery (79 pts) followed by chemotherapy ** (69 pts)	87.5%	25	39%
		Chemotherapy ** (40 pts) followed by surgery			
Karrer K. et al., 1995 [29]	183	Surgery followed by Chemotherapy ***	83%	30	n.r.
Fujimori K. et al., 1997 [30]	22	Chemotherapy followed by surgery ****	95.5%	61.9	50%
Rea F. et al., 1998 [31]	104	Surgery followed by chemotherapy + radiotherapy (51 pts)	100%	28	32%
		Chemotherapy followed by surgery + radiotherapy (53 pts)			
Eberhardt W. et al., 1999 [32]	46	Chemotherapy ^ ± RT followed by surgery	72%	36	46%
Tsuchiya R. et al., 2005 [33]	61	Surgery followed by chemotherapy ^	100%	Not reached	57%

OS: overall survival; n.r.: not reported; pts: patients; RT: radiotherapy; * CAV (cyclophosphamide, doxorubicin and vincristine) or cisplatin + etoposide; ** CAV or CAV + etoposide or CAV + methotrexate; *** CAV or cyclophaosphamide, lomustine, methotrexate/CAV/ifosfamide + mesna + etoposide; **** CAV II (cyclophosphamide, doxorubicin and etoposide) or PE (cisplatin + etoposide); ^ PE (cisplatin + etoposide).

**Table 3 cancers-13-00390-t003:** Retrospective trials evaluating the role of surgery.

Author	Patients	Treatment	Complete Resection Rate (%)	Median OS (Months)	5-Year OS
Shields T.W. et al., 1982 [34]	132	Surgery followed by adjuvant chemotherapy	100%	n.r.	23% (59.9% T1N0)
Osterlind K. et al., 1986 [35]	52	Surgery followed by adjuvant chemotherapy	69.2%	30	16% (at 30 months)
Hara N. et al., 1991 [36]	36	Surgery followed by chemotherapy (19 pts)	44.4%	33	38%
		Chemotherapy followed by surgery (17 pts)			
Inoue M. et al., 2000 [37]	91	Surgery followed by chemotherapy (71 pts) or radiotherapy (17 pts)	89%	26	37.1%
Badzio A. et al., 2004 [38]	67	Surgery followed by adjuvant chemotherapy	100%	22	27%
Brock M.V. et al., 2005 [39]	82	Surgery alone (9 pts)	96.3%	24	42%
		Chemotherapy followed by surgery (18 pts)			
		Surgery followed by chemotherapy (45 pts) or other therapy (10 pts)			
Takenaka T. et al., 2015 [40]	88	Surgery alone (16 pts)	n.r.	18	59% Stage I
		Surgery + chemotherapy (63 pts)			39% Stage II
		Surgery + chemoradiotherapy (9 pts)			14% Stage III
Zhong L. et al., 2020 [41]	50	Surgery followed by chemotherapy and radiotherapy (30 pts)	n.r.	79	28%
		Chemotherapy followed by surgery, chemotherapy ± radiotherapy (20 pts)			
Casiraghi M. et al., 2020 [42]	65	Surgery upfront (39 pts) followed by chemotherapy	100%	36	42% (76.6% Stage I)
		Chemotherapy followed by surgery (26 pts)			

OS: overall survival; n.r.: not reported; pts: patients.

**Table 4 cancers-13-00390-t004:** Cancer database review.

Author	Database	All SCLC Patients	Stage I SCLC	Treatment	Resection Rate (%)	Median OS (Months)	5-Year OS
Rostad H. et al., 2004 [43]	Norway Cancer Registry	2442	180	Surgery (38 pts) followed by chemotherapy or radiotherapy (25 pts)	21%	54	44.9%
Schreiber D. et al., 2010 [8]	SEER	14,179	2382	Surgery (863 pts) followed by radiotherapy (241 pts)	36%	28	34.6%
Yu JB. et al., 2010 [7]	SEER	1560	1560	Surgery (378 pts) followed by radiotherapy (38 pts)	24.2%	58	50.3% (surgery alone)
							57.1 (surgery + RT)
Varlotto JM et al., 2011 [44]	SEER	2214	1690	Surgery (448 pts) followed by radiotherapy (59 pts)	26.5%	50	47.4% (surgery alone)
Weksler B. et al., 2012 [45]	SEER	3566	2686	Surgery (683 pts) followed by radiotherapy (202 pts)	25.4%	34	29.6%
Gaspar LE. et al., 2012 [46]	NCDB	68,611	4103	Surgery (1395 pts)	34%	30.8	9.7% (LS-SCLC)
Takei H. et al., 2014 [47]	Japanese Lung Cancer Registry	243	168	Surgery (168 pts) followed by chemotherapy (158 pts)	88.1% *	Not reached	52.6%
Yang CGJ. et al., 2017 [48]	NCDB	4490	1041 °	Surgery (96 pts) followed by chemotherapy ± radiotherapy	9.2%	33.3	31.4%
Combs SE. et al., 2015 [50]	NCDB	203,229	4893	Surgery (1009 pts) followed by chemotherapy	20.6%	Not reached	51%

SEER: Surveillance, Epidemiology, and End Results database; NCDB: National Cancer Data Base; n.r.: not reported; RT: radiotherapy; pts: patients; LS-SCLC: limited-stage Small-Cell Lung Cancer; * Complete resection rate (R0); ° Stage I–II; pts: patients.

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
