# Peer review of "Surgery in Small-Cell Lung Cancer"

_cancers, 2021, doi:10.3390/cancers13030390_

Round 1

Reviewer 1 Report

The manuscript presents a literature review about the main results of surgery in SCLC. The review summarizes the existing studies from two aspects: only surgery as a treatment for SCLC and surgery plus chemotherapy/radiotherapy in SCLC.  The manuscript is well-organized, however, there are a few major comments that should be considered.

  • The contribution of the review is limited. The authors just summarized the existing studies. As a suggestion, some quantitative comparative analysis of the collected literature results could be added.
  • I would suggest the authors summarizes the contributions and list them in the introduction section.
  • As a literature review, the assumption of what and how the literature was selected is missing in the manuscript.
  • The scope of the review, such as time range, problem categories, was not defined clearly.
  • The current manuscript doesn’t include any tables or figures. It is a little challenging for the reader to follow the content, such as the survival rate in different studies. I would suggest having tables or figures to summarize the key results from reviewed literature, such as patient population, study approach, survival rate, conclusion, etc.

Reviewer 2 Report

Thank you for giving me the opportunity to review this manuscript. English is not my first language, so I am not able to correct grammar mistakes (if present).

This article is a review article about the role of surgery in small cell lung carcinoma of the lung. This review of the literature provides an exhaustive review of all published articles in the field. The studies are classified according to their design. The review not only focusses on recent articles but also focuses on articles published in the 60s.

This review is well written and deserves publication. Nevertheless, the information showed to the reader are not new, and the role of surgery is not a “hot topic” in the field. Several recent reviews of the literature have also been published on the same topic in other journals and do not see what is new in this literature review.

I think that the authors should provide tables summarizing the main studies discussed in the text. I think it would make the manuscript easier to read for readers.

Round 2

Reviewer 1 Report

All the comments are well addressed. Would recommend for acceptance. Thank you.